# FlexMatch: Boosting Semi-Supervised Learning with Curriculum Pseudo Labeling

**Bowen Zhang**[*]
Tokyo Institute of Technology
bowen.z.ab@m.titech.ac.jp

**Yidong Wang**[*]
Tokyo Institute of Technology
wang.y.ca@m.titech.ac.jp

**Wenxin Hou**
Microsoft
wenxinhou@microsoft.com

**Hao Wu**
Tokyo Institute of Technology
wu.h.aj@m.titech.ac.jp

**Jindong Wang**[†]
Microsoft Research Asia
jindwang@microsoft.com

**Manabu Okumura**[†]
Tokyo Institute of Technology
oku@pi.titech.ac.jp

**Takahiro Shinozaki**[†]
Tokyo Institute of Technology
shinot@ict.e.titech.ac.jp

## Abstract

The recently proposed FixMatch achieved state-of-the-art results on most semi-supervised learning (SSL) benchmarks. However, like other modern SSL algorithms, FixMatch uses a pre-defined constant threshold for all classes to select unlabeled data that contribute to the training, thus failing to consider different learning status and learning difficulties of different classes. To address this issue, we propose Curriculum Pseudo Labeling (CPL), a curriculum learning approach to leverage unlabeled data according to the model's learning status. The core of CPL is to flexibly adjust thresholds for different classes at each time step to let pass informative unlabeled data and their pseudo labels. CPL does not introduce additional parameters or computations (forward or backward propagation). We apply CPL to FixMatch and call our improved algorithm *FlexMatch*. FlexMatch achieves state-of-the-art performance on a variety of SSL benchmarks, with especially strong performances when the labeled data are extremely limited or when the task is challenging. For example, FlexMatch achieves **13.96%** and **18.96%** error rate reduction over FixMatch on CIFAR-100 and STL-10 datasets respectively, when there are only 4 labels per class. CPL also significantly boosts the convergence speed, e.g., FlexMatch can use only 1/5 training time of FixMatch to achieve even better performance. Furthermore, we show that CPL can be easily adapted to other SSL algorithms and remarkably improve their performances. We open-source our code at https://github.com/TorchSSL/TorchSSL.

## 1   Introduction

Semi-supervised learning (SSL) has attracted increasing attention in recent years due to its superiority in leveraging a large amount of unlabeled data. This is particularly advantageous when the labeled data are limited in quantity or laborious to obtain. Consistency regularization [1–3] and pseudo labeling [4–8] are two powerful techniques for utilizing unlabeled data and have been widely used in modern SSL algorithms [9–13]. The recently proposed FixMatch [14] achieves competitive results

---

[*]Equal contribution.
[†]Corresponding author.

by combining these techniques with weak and strong data augmentations and using cross-entropy loss as the consistency regularization criterion.

However, a drawback of FixMatch and other popular SSL algorithms such as Pseudo-Labeling [4] and Unsupervised Data Augmentation (UDA) [11] is that they rely on a *fixed* threshold to compute the unsupervised loss, using only unlabeled data whose prediction confidence is above the threshold. While this strategy can make sure that only high-quality unlabeled data contribute to the model training, it ignores a considerable amount of other unlabeled data, especially at the early stage of the training process, where only a few unlabeled data have their prediction confidence above the threshold. Moreover, modern SSL algorithms handle all classes *equally* without considering their different learning difficulties.

To address these issues, we propose Curriculum Pseudo Labeling (CPL), a curriculum learning [15] strategy to take into account the learning status of each class for semi-supervised learning. CPL substitutes the pre-defined thresholds with *flexible* thresholds that are dynamically adjusted for each class according to the current learning status. Notably, this process does not introduce any additional parameter (hyper-parameter or trainable parameter) or extra computation (forward or back propagation). We apply this curriculum learning strategy directly to FixMatch and call the improved algorithm *FlexMatch*.

While the training speed remains as efficient as that of FixMatch, FlexMatch converges significantly faster and achieves state-of-the-art performances on most SSL image classification benchmarks. The benefit of introducing CPL is particularly remarkable when the labels are scarce or when the task is challenging. For instance, on the STL-10 dataset, FlexMatch achieves relative performance improvement over FixMatch by 18.96%, 16.11%, and 7.68% when the label amount is 400, 2500, and 10000 respectively. Moreover, CPL further shows its superiority by boosting the convergence speed – with CPL, FlexMatch takes less than 1/5 training time of FixMatch to reach its final accuracy. Adapting CPL to other modern SSL algorithms also leads to improvements in accuracy and convergence speed.

To sum up, this paper makes the following three contributions:

- We propose Curriculum Pseudo Labeling (CPL), a curriculum learning approach of dynamically leveraging unlabeled data for SSL. It is almost cost-free and can be easily integrated to other SSL methods.

- CPL significantly boosts the accuracy and convergence performance of several popular SSL algorithms on common benchmarks. Specifically, FlexMatch, the integration of FixMatch and CPL, achieves state-of-the-art results.

- We open-source TorchSSL, a unified PyTorch-based semi-supervised learning codebase for the fair study of SSL algorithms. TorchSSL includes implementations of popular SSL algorithms and their corresponding training strategies, and is easy to use or customize.

## 2   Background

Consistency regularization follows the continuity assumption of SSL [1, 2]. The most basic consistency loss in SSL, such as in $\Pi$ Model [9], Mean Teacher [10] and MixMatch [12], is the $\ell$-2 loss:

$$\sum_{b=1}^{\mu B} ||p_m(y|\omega(u_b)) - p_m(y|\omega(u_b))||_2^2,  \tag{1}$$

where $B$ is the batch size of labeled data, $\mu$ is the ratio of unlabeled data to labeled data, $\omega$ is a stochastic data augmentation function (thus the two terms in Eq.(1) are different), $u_b$ denotes a piece of unlabeled data, and $p_m$ represents the output probability of the model. With the introduction of pseudo labeling techniques [5, 7], the consistency regularization is converted to an entropy minimization process [16], which is more suitable for the classification task. The improved consistency loss with pseudo labeling can be represented as:

$$\frac{1}{\mu B} \sum_{b=1}^{\mu B} \mathbb{1}(\max(p_m(y|\omega(u_b))) > \tau) H(\hat{p}_m(y|\omega(u_b)), p_m(y|\omega(u_b))),  \tag{2}$$

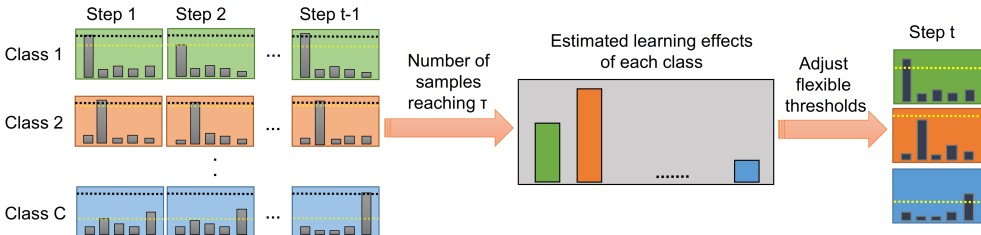

Figure 1: Illustration of Curriculum Pseudo Label (CPL). The estimated learning effects of each class are decided by the number of unlabeled data samples falling into this class and above the fixed threshold. They are then used to adjust the flexible thresholds to let pass the optimal unlabeled data. Note that the estimated learning effects do not always grow – they may also decrease if the predictions of the unlabeled data fall into other classes in later iterations.

where $H$ is cross-entropy, $\tau$ is the pre-defined threshold and $\hat{p}_m(y|\omega(u_b))$ is the pseudo label that can either be a 'hard' one-hot label [4, 14] or a sharpened 'soft' one [11]. The intention of using a threshold is to mask out noisy unlabeled data that have low prediction confidence.

FixMatch utilizes such consistency regularization with strong augmentation to achieve competitive performance. For unlabeled data, FixMatch first uses weak augmentation to generate artificial labels. These labels are then used as the target of strongly-augmented data. The unsupervised loss term in FixMatch thereby has the form:

$$\frac{1}{\mu B} \sum_{b=1}^{\mu B} \mathbb{1}(\max(p_m(y|\omega(u_b))) > \tau) H(\hat{p}_m(y|\omega(u_b)), p_m(y|\Omega(u_b))), \tag{3}$$

where $\Omega$ is a strong augmentation function instead of weak augmentation $\omega$.

Of the aforementioned works, the pre-defined threshold ($\tau$) is constant. We believe this can be improved because the data of some classes may be inherently more difficult to learn than others. Curriculum learning [15] is a learning strategy where learning samples are gradually introduced according to the model's learning process. In such a way, the model is always optimally challenged. This technique is widely employed in deep learning research [17–21].

## 3 FlexMatch

### 3.1 Curriculum Pseudo Labeling

While current SSL algorithms render pseudo labels of only high-confidence unlabeled data cut off by a pre-defined threshold, CPL renders the pseudo labels *to different classes* and *at different time steps*. Such a process is realized by adjusting the thresholds according to the model's learning status of each class.

However, it is non-trivial to dynamically determine the thresholds according to the learning status. The most ideal approach would be calculating evaluation accuracies for each class and use them to scale the threshold, as:

$$\mathcal{T}_t(c) = a_t(c) \cdot \tau, \tag{4}$$

where $\mathcal{T}_t(c)$ is the flexible threshold for class $c$ at time step $t$ and $a_t(c)$ is the corresponding evaluation accuracy. In this way, lower accuracy that indicates a less satisfactory learning status of the class will lead to a lower threshold that encourages more samples of this class to be learned. Since we cannot use the evaluation set in the model learning process, one may have to separate an extra validation set from the training set for such accuracy evaluations. However, this practice show two fatal problems: First, such a *labeled* validation set separated from the training set is expensive under SSL scenario as the labeled data are already scarce. Second, to dynamically adjust the thresholds in the training process, accuracy evaluations must be done continually at each time step $t$, which will considerably slow down the training speed.

In this work, we propose Curriculum Pseudo Labeling (CPL) for semi-supervised learning. Our CPL uses an alternative way to estimate the learning status, which does not introduce additional inference processes, nor needs an extra validation set. As believed in [14], a high threshold that filters out noisy pseudo labels and leaves only high-quality ones can considerably reduce the confirmation bias [22]. Therefore, our key assumption is that when the threshold is high, the learning effect of a class can be reflected by the number of samples whose predictions fall into this class and above the threshold. Namely, the class with fewer samples having their prediction confidence reach the threshold is considered to have a greater learning difficulty or a worse learning status, formulated as:

$$\sigma_t(c) = \sum_{n=1}^{N} \mathbb{1}(\max(p_{m,t}(y|u_n)) > \tau) \cdot \mathbb{1}(\arg\max(p_{m,t}(y|u_n) = c). \tag{5}$$

where $\sigma_t(c)$ reflects the learning effect of class $c$ at time step $t$. $p_{m,t}(y|u_n)$ is the model's prediction for unlabeled data $u_n$ at time step $t$, and $N$ is the total number of unlabeled data. When the unlabeled dataset is balanced (i.e., the number of unlabeled data belonging to different classes are equal or close), larger $\sigma_t(c)$ indicates a better estimated learning effect. By applying the following normalization to $\sigma_t(c)$ to make its range between 0 to 1, it can then be used to scale the fixed threshold $\tau$:

$$\beta_t(c) = \frac{\sigma_t(c)}{\max\limits_{c} \sigma_t}, \tag{6}$$

$$\mathcal{T}_t(c) = \beta_t(c) \cdot \tau. \tag{7}$$

One characteristic of such a normalization approach is that the best-learned class has its $\beta_t(c)$ equal to 1, causing its flexible threshold equal to $\tau$. This is desirable. For classes that are hard to learn, the thresholds are lowered down, encouraging more training samples in these classes to be learned. This also improves the data utilization ratio. As learning proceeds, the threshold of a well-learned class is raised higher to selectively pick up higher-quality samples. Eventually, when all classes have reached reliable accuracies, the thresholds will all approach $\tau$. Note that the thresholds do not always grow, it may also decrease if the unlabeled data is classified into a different class in later iterations. This new threshold is used for calculating the unsupervised loss in FlexMatch, which can be formulated as:

$$\mathcal{L}_{u,t} = \frac{1}{\mu B} \sum_{b=1}^{\mu B} \mathbb{1}(\max(q_b) > \mathcal{T}_t(\arg\max(q_b))) H(\hat{q}_b, p_m(y|\Omega(u_b))), \tag{8}$$

where $q_b = p_m(y|\omega(u_b))$. The flexible thresholds are updated at each iteration. Finally, we can formulate the loss in FlexMatch as the weighted combination (by $\lambda$) of supervised and unsupervised loss:

$$\mathcal{L}_t = \mathcal{L}_s + \lambda \mathcal{L}_{u,t}, \tag{9}$$

where $\mathcal{L}_s$ is the supervised loss on labeled data:

$$\mathcal{L}_s = \frac{1}{B} \sum_{b=1}^{B} H(y_b, p_m(y|\omega(x_b))). \tag{10}$$

Note that the cost of introducing CPL is almost free. Practically, every time the prediction confidence of an unlabeled data $u_n$ is above the fixed threshold $\tau$, the data, and its predicted class are marked and will be used for calculating $\beta_t(c)$ at the next time step. Such marking actions are bonus actions each time the consistency loss is computed. Therefore, FlexMatch does not introduce additional forward propagation processes for evaluating the model's learning status, nor new parameters.

## 3.2 Threshold warm-up

We noticed in our experiments that at the early stage of the training, the model may blindly predict most unlabeled samples into a certain class depending on the parameter initialization(i.e., more likely to have confirmation bias). Hence, the estimated learning status may not be reliable at this stage. Therefore, we introduce a warm-up process by rewriting the denominator in Eq. (6) as:

$$\beta_t(c) = \frac{\sigma_t(c)}{\max\left\{\max\limits_{c} \sigma_t, N - \sum\limits_{c} \sigma_t\right\}}, \tag{11}$$

**Algorithm 1** FlexMatch algorithm.

---

1: **Input:** $\mathcal{X} = \{(x_m, y_m) : m \in (1, \ldots, M)\}, \ \mathcal{U} = \{u_n : n \in (1, \ldots, N)\}$ {M labeled data and N unlabeled data.}
2: $\hat{u}_n = -1 : n \in (1, \ldots, N)$ {Initialize predictions of all unlabeled data as -1 indicating unused.}
3: **while** not reach the maximum iteration **do**
4:    **for** $c = 1$ to $C$ **do**
5:       $\sigma(c) = \sum_{n=1}^{N} \mathbb{1}(\hat{u}_n = c)$ {Compute estimated learning effect.}
6:       **if** $\max \sigma(c) < \sum_{n=1}^{N} \mathbb{1}(\hat{u}_n = -1)$ **then**
7:          Calculate $\beta(c)$ using Eq. (11) {Threshold warms up when unused data dominate.}
8:       **else**
9:          Calculate $\beta(c)$ using Eq. (6) {Compute normalized estimated learning effect.}
10:      **end if**
11:      Calculate $\mathcal{T}(c)$ using Eq. (7) {Determine the flexible threshold for class $c$.}
12:    **end for**
13:    **for** $b = 1$ to $\mu B$ **do**
14:       **if** $p_m(y|\omega(u_b)) > \tau$ **then**
15:          $\hat{u}_b = \arg \max q_b$ {Update the prediction of unlabeled data $u_b$.}
16:      **end if**
17:    **end for**
18:    Compute the loss via Eq. (8), (10) and (9).
19: **end while**
20: **Return:** Model parameters.

---

where the term $N - \sum_{c=1}^{C} \sigma_t(c)$ can be regarded as the number of unlabeled data that have not been used. This ensures that at the beginning of the training, all estimated learning effects gradually rise from 0 until the number of unused unlabeled data is no longer predominant. The duration of such a period depends on the unlabeled data amount (ref. $N$ in Eq. (11)) and the learning difficulty (ref. the growing speed of $\sigma_t(c)$ in Eq. (11)) of the dataset. In practice, such a warm-up process is very easy to implement as we can add an extra class to denote the unused unlabeled data. Thus calculating the denominator of Eq. (11) is simply converted to finding the maximum among $c + 1$ classes.

### 3.3 Non-linear mapping function

The flexible threshold in Eq. (7) is determined by the normalized estimated learning effects via a linear mapping. However, it may not be the most suitable mapping in the real training process, where the increase or decrease of $\beta_t(c)$ may make big jumps in the early phase where the predictions of the model are still unstable; and only make small fluctuations after the class is well-learned in the mid and late training stage. Therefore, it is preferable if the flexible thresholds can be more sensitive when $\beta_t(c)$ is large and vice versa.

We propose a non-linear mapping function to enable the thresholds to have a non-linear increasing curve when $\beta_t(c)$ ranges uniformly from 0 to 1, as formulated below:

$$\mathcal{T}_t(c) = \mathcal{M}(\beta_t(c)) \cdot \tau, \tag{12}$$

where $\mathcal{M}(\cdot)$ is a non-linear mapping function. It is clear that Eq. (7) can be seen as a special case by setting $\mathcal{M}$ to the identity function. The mapping function $\mathcal{M}$ should be monotonically increasing and have a maximum no larger than $1/\tau$ (otherwise the flexible threshold can be larger than 1 and filter out all samples). To avoid introducing additional hyper-parameters (e.g. lower limits of the flexible thresholds), we consider the mapping function to have a range from 0 to 1 so that the flexible thresholds range from 0 to $\tau$.

A monotone increasing convex function lets the thresholds grow slowly when $\beta_t(c)$ is small, and become more sensitive as $\beta_t(c)$ gets larger. Hence, we intuitively choose a convex function with the above-mentioned properties $\mathcal{M}(x) = \frac{x}{2-x}$ for our experiments. We also conduct an ablation study to compare among mapping functions with different convexity and concavity in Sec. 4.4. The full algorithm of FlexMatch is shown in Algorithm 1.

Table 1: Error rates on CIFAR-10/100, SVHN, and STL-10 datasets. The 'Flex' prefix denotes applying CPL to the algorithm, and 'PL' is an abbreviation of Pseudo-Labeling. STL-10 dataset does not have label information for unlabeled data, thus its fully-supervised result is unavailable.

| Dataset | CIFAR-10 | | | CIFAR-100 | | | STL-10 | | | SVHN | |
|---|---|---|---|---|---|---|---|---|---|---|---|
| Label Amount | 40 | 250 | 4000 | 400 | 2500 | 10000 | 40 | 250 | 1000 | 40 | 1000 |
| PL | $74.61_{\pm0.26}$ | $46.49_{\pm2.20}$ | $15.08_{\pm0.19}$ | $87.45_{\pm0.85}$ | $57.74_{\pm0.28}$ | $36.55_{\pm0.24}$ | $74.68_{\pm0.99}$ | $55.45_{\pm2.43}$ | $32.64_{\pm0.71}$ | $64.61_{\pm5.60}$ | $\mathbf{9.40}_{\pm0.32}$ |
| Flex-PL | $\mathbf{73.74}_{\pm1.96}$ | $\mathbf{46.14}_{\pm1.81}$ | $\mathbf{14.75}_{\pm0.19}$ | $\mathbf{85.72}_{\pm0.46}$ | $\mathbf{56.12}_{\pm0.51}$ | $\mathbf{35.60}_{\pm0.15}$ | $\mathbf{73.42}_{\pm2.19}$ | $\mathbf{52.06}_{\pm2.50}$ | $\mathbf{32.05}_{\pm0.37}$ | $\mathbf{63.21}_{\pm3.64}$ | $12.05_{\pm0.54}$ |
| UDA | $10.62_{\pm3.75}$ | $5.16_{\pm0.06}$ | $4.29_{\pm0.07}$ | $46.39_{\pm1.59}$ | $27.73_{\pm0.21}$ | $22.49_{\pm0.23}$ | $37.42_{\pm8.44}$ | $9.72_{\pm1.15}$ | $6.64_{\pm0.17}$ | $5.12_{\pm4.27}$ | $\mathbf{1.89}_{\pm0.01}$ |
| Flex-UDA | $\mathbf{5.44}_{\pm0.52}$ | $\mathbf{5.02}_{\pm0.07}$ | $\mathbf{4.24}_{\pm0.06}$ | $\mathbf{45.17}_{\pm1.88}$ | $\mathbf{27.08}_{\pm0.15}$ | $\mathbf{21.91}_{\pm0.10}$ | $\mathbf{29.53}_{\pm2.10}$ | $\mathbf{9.03}_{\pm0.45}$ | $\mathbf{6.10}_{\pm0.25}$ | $\mathbf{3.42}_{\pm1.51}$ | $2.02_{\pm0.05}$ |
| FixMatch | $7.47_{\pm0.28}$ | $\mathbf{4.86}_{\pm0.05}$ | $4.21_{\pm0.08}$ | $46.42_{\pm0.82}$ | $28.03_{\pm0.16}$ | $22.20_{\pm0.12}$ | $35.97_{\pm4.14}$ | $9.81_{\pm1.04}$ | $6.25_{\pm0.33}$ | $\mathbf{3.81}_{\pm1.18}$ | $\mathbf{1.96}_{\pm0.03}$ |
| FlexMatch | $\mathbf{4.97}_{\pm0.06}$ | $4.98_{\pm0.09}$ | $\mathbf{4.19}_{\pm0.01}$ | $\mathbf{39.94}_{\pm1.62}$ | $\mathbf{26.49}_{\pm0.20}$ | $\mathbf{21.90}_{\pm0.15}$ | $\mathbf{29.15}_{\pm4.16}$ | $\mathbf{8.23}_{\pm0.39}$ | $\mathbf{5.77}_{\pm0.18}$ | $8.19_{\pm3.20}$ | $6.72_{\pm0.30}$ |
| Fully-Supervised | | $4.62_{\pm0.05}$ | | | $19.30_{\pm0.09}$ | | | - | | | $2.13_{\pm0.02}$ | |

## 4  Experiments

We evaluate FlexMatch and other CPL-enabled algorithms on common SSL datasets: CIFAR-10/100 [23], SVHN [24], STL-10 [25] and ImageNet [26], and extensively investigate the performance under various labeled data amounts. We mainly compare our method with Pseudo-Labeling [4], UDA [11] and FixMatch [14], since they all involve a pre-defined threshold. The results of other popular SSL algorithms are in the appendix. We also add a fully-supervised experiment for each dataset to better understand the results of SSL algorithms. Note that previously suggested [27] fully-supervised comparisons use only the labeled set for training, whose purpose is to manifest the improvement brought by the introduction of unlabeled data. With the development of modern SSL algorithms, however, semi-supervised approaches are achieving competitive performance with supervised ones, or even better performance due to the strength of consistency regularization. Therefore, our fully-supervised comparisons are conducted with all data labeled, and apply weak data augmentations following Eq. (10). We re-implement all baselines using our PyTorch [28] codebase: TorchSSL, which is introduced in the appendix.

For a fair comparison, we use the same hyper-parameters following FixMatch [14]. Concretely, the optimizer for all experiments is standard stochastic gradient descent (SGD) with a momentum of 0.9 [29, 30]. For all datasets, we use an initial learning rate of $0.03$ with a cosine learning rate decay schedule [31] as $\eta = \eta_0 \cos(\frac{7\pi k}{16K})$, where $\eta_0$ is the initial learning rate, $k$ is the current training step and $K$ is the total training step that is set to $2^{20}$. We also perform an exponential moving average with the momentum of $0.999$. The batch size of labeled data is $64$ except for ImageNet. $\mu$ is set to be 1 for Pseudo-Label and 7 for UDA, FixMatch, and FlexMatch. $\tau$ is set to 0.8 for UDA and 0.95 for Pseudo Label, FixMatch, and FlexMatch. These setups follow the original papers. The strong augmentation function used in our experiments is RandAugment [32]. We use ResNet-50 [33] for the ImageNet experiment and Wide ResNet (WRN) [34] and its variant [35] for other datasets. Detailed hyper-parameters are listed in the appendix.

We adopt two evaluation metrics: (1) the median error rate of the last 20 checkpoints following [12, 14], and (2) the best error rate in all checkpoints. We argue that the median approach is not suitable when the convergence speeds of the algorithms show significant differences – the large number of redundant iterations may result in over-fitting for the fast-converge algorithms. Therefore, we report the best error rates for all algorithms, while the results of the median approach are also provided in the appendix, showing that our FlexMatch still achieves the best performance. We run each task three times using distinct random seeds to obtain the error bars.

### 4.1  Main results

The classification error rates on CIFAR-10/100, STL-10 and SVHN datasets are in Table 1, and the results on ImageNet are in Sec. 4.2. Note that the SVHN dataset used in our experiment also includes the extra set that contains 531,131 additional samples. Results demonstrate that FlexMatch achieves the state-of-the-art performance on most of the benchmark datasets except for SVHN where Flex-UDA (i.e., UDA with CPL) and UDA have the lowest error rate on the 40-label split and the 1000-label split, respectively. We also provide the detailed precision, recall, F1, and AUC results in the appendix. Our CPL (FlexMatch) has the following advantages:

**CPL achieves better performance on tasks with extremely limited labeled data.**  Our FlexMatch significantly outperforms other methods when the amount of labels is extremely small. For instance,

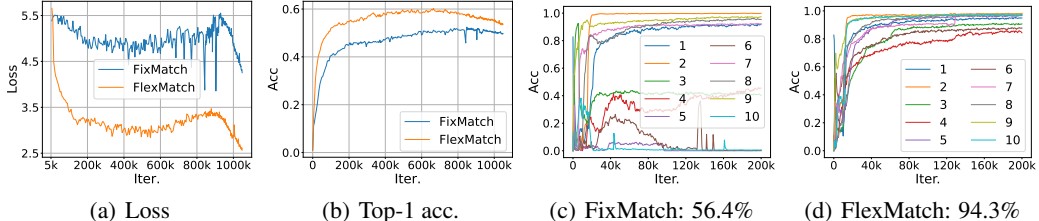

| (a) Loss | (b) Top-1 acc. | (c) FixMatch: 56.4% | (d) FlexMatch: 94.3% |

Figure 3: Convergence analysis of FixMatch and FlexMatch. (a) and (b) depict the loss and top-1-accuracy on CIFAR-100 with 400 labels. Evaluations are done every 5K iterations. (c) and (d) demonstrate the class-wise accuracy within the first 200K iterations on CIFAR-10 dataset. The numbers in legend correspond to the ten classes in the dataset.

on the CIFAR-100 dataset with 400 labels (i.e., only 4 label samples per class), FlexMatch achieves an average error rate of **39.94**%, which significantly outperforms FixMatch (46.42%).

**CPL improves the performance of existing SSL algorithms.** Other than FixMatch, CPL can also improve the performance of other existing SSL algorithms such as Pseudo-Labeling and UDA. For instance, the error rate is reduced from 37.4% to 29.53% for UDA on the STL-10 40-label split after introducing CPL (refer to as Flex-UDA in Table 1). These results further prove the effectiveness of CPL in better leveraging unlabeled data. Figure 2 shows the average running time of a single iteration with or without adding our CPL, it is clear that while improving the performance of existing SSL algorithms, our CPL *does not* introduce additional computational burden.

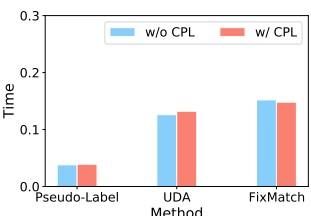

Figure 2: Average running time of one iteration on a single GeForce RTX 3090 GPU.

**CPL achieves better performance on complicated tasks.** The STL-10 dataset contains unlabeled data from a similar but broader distribution of images than its labeled set. The existence of new types of objects in the unlabeled dataset makes STL-10 a more challenging and realistic task. FlexMatch achieves greater performance improvement under such a challenging situation. The error rate on STL-10 with only 40 labels is 29.15%, which is relatively **18.96**% better than FixMatch (35.97%). Similar strong improvements are also observed on CIFAR-100 dataset, which has as many as one hundred classes.

We also analyze the reason why FlexMatch performs less favorably on SVHN. This is probably because SVHN is a relatively *simple* (i.e., to classify digits) yet *unbalanced* dataset. The class-wise imbalance leads to the classes with fewer samples never have their estimated learning effects close to 1 according to Eq. (6), even when they are already well-learned. Such low thresholds allow noisy pseudo-labeled samples to be trusted and learned throughout the training process, which is also reflected by the loss descent curve where the low-threshold classes have major fluctuations. FixMatch, on the other hand, fixes its threshold at 0.95 to filter out noisy samples. Such a fixed high threshold is not preferable with respect to both accuracies of hard-to-learn classes and overall convergence speed as explained earlier, but since SVHN is an easy task, the model can easily learn the task and make high-confidence predictions, setting a high-fixed threshold thus becomes less problematic and has its advantages overweighed.

## 4.2 Results on ImageNet

We also verify the effectiveness of CPL on ImageNet-1K [26] which is a much more realistic and complicated dataset. We randomly choose the same 100K labeled data (i.e., 100 labels per class), which is less than $8\%$ of the total labels. The hyper-parameters used for ImageNet can be found in the appendix, where the two algorithms share the same hyper-parameters. We show the error rate comparison after running $2^{20}$ iterations in Table 2. This result indicates that when the task is complicated, despite the class imbalance issue (the number

Table 2: Error rate results on ImageNet after $2^{20}$ iterations.

| Method | Top-1 | Top-5 |
|---|---|---|
| FixMatch | 43.66 | 21.80 |
| FlexMatch | **42.02** | **19.49** |

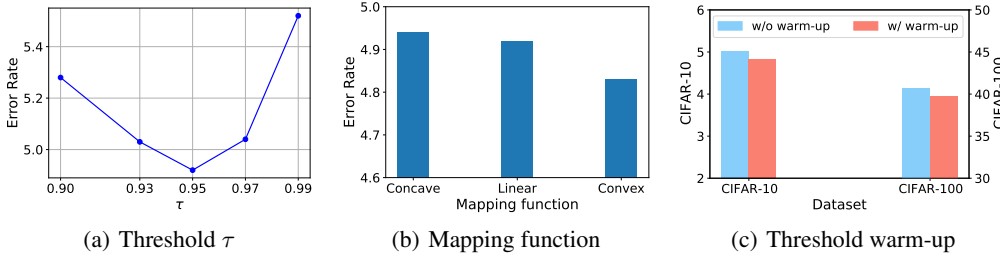

| (a) Threshold $\tau$ | (b) Mapping function | (c) Threshold warm-up |

Figure 4: Ablation study of FlexMatch.

of images within each class ranges from 732 to 1300), CPL can still bring improvements. Note that this result does not represent the best performance of each algorithm as the model cannot fully converge after $2^{20}$ iterations, and due to the computational resource limitation, we did not further tune the hyper-parameters to obtain the best results on ImageNet.

### 4.3 Convergence speed acceleration

Another strong advantage of FlexMatch is its superior convergence speed. Figure 3(a) and 3(b) shows the comparison between FlexMatch and FixMatch with respect to the loss and top-1-accuracy on CIFAR-100 400-label split. The loss of FlexMatch decreases much faster and smoother than FixMatch, demonstrating its superior convergence speed. The major fluctuations of the loss in FixMatch may due to the pre-defined threshold that lets pass most unlabeled data belonging to certain classes, whereas with CPL a larger batch of unlabeled data containing samples from various classes enables the gradient to more directly head toward the global optimum. As a result, with only 50K iterations, FlexMatch has already surpassed the final results of FixMatch. After 800K iterations, however, we observe a further decrease in loss and accuracy. This is likely due to over-fitting, which also occurs in FixMatch after 900K iterations. Thus, we believe it is not fair to use the median results of the last few checkpoints for evaluating algorithms with different convergence speeds.

We further compare the class-wise accuracy of FixMatch and FlexMatch on CIFAR-10 in their early training stages. As shown in Figure 3(c) and 3(d), at iteration 200K, FixMatch only hits an overall accuracy of 56.35% as half of the classes are still learned unsatisfactorily, whereas FlexMatch has already achieved an overall accuracy of 94.29% which is even higher than the final accuracy reached by FixMatch after 1M iterations. It is manifest that the introduction of CPL successfully encourages the model to proactively learn those difficult classes thereby improving the overall learning effect.

### 4.4 Ablation study

We conduct experiments to evaluate three components of FlexMatch: the upper limit of thresholds $\tau$, mapping functions $\mathcal{M}(x)$, and threshold warm-up.

**Threshold upper bound.** We investigate 5 different $\tau$ values and 3 different mapping functions on CIFAR-10 dataset with 40 labels. As shown in Figure 4(a), the optimal choice of $\tau$ is around 0.95, either increasing or decreasing this value results in a performance decay. Note that in FlexMatch, tuning $\tau$ does not only affect the upper limit of the threshold but also the estimated learning effects because they are determined by the number of samples that fall above $\tau$.

**Mapping function.** We explore three different mapping functions in Figure 4(b): (1) concave: $\mathcal{M}(x) = \ln(x + 1)/\ln 2$, (2) linear: $\mathcal{M}(x) = x$, and (3) convex: $\mathcal{M}(x) = x/(2 - x)$. We see that the convex function shows the best performance and the concave function shows the worst. Although tweaking the degree of convexity may probably lead to further improvement, we do not make further investigation in this paper. It is noteworthy that all these functions have their outputs grow from 0 to 1 when the inputs go from 0 to 1. One may also design a function with a different range, for instance, from 0.5 to 1. In this case, it is equivalent to setting a lower limit to the flexible threshold so that even at the beginning of the training, only samples with prediction confidence higher than this limit will contribute to the unsupervised loss. We do not include such a lower limit in FlexMatch since it will introduce a new hyper-parameter. However, we did find that setting a lower limit at 0.5 can slightly

improve the performance. A possible reason is that the lower threshold prevents noisy training caused by incorrect pseudo labels at the early stage [36].

**Threshold warm-up.**   We analyze the performance of threshold warm-up on both CIFAR-10 (40 labels) and CIFAR-100 (400 labels) datasets. As shown in Figure 4(c), threshold warm-up can bring about 0.2% absolute improvement on CIFAR-10 and about 1% on CIFAR-100. At the beginning of the training without the threshold warm-up, the flexible thresholds may go through heavy fluctuations because the denominator in Eq.(6) is small. In the meantime, there will always be some classes whose flexible thresholds reach or approach $\tau$, thereby filtering out most unlabeled data in the batch. The threshold warm-up solves this issue by gradually raising the thresholds of all classes from zero – it creates a learning boom at the early training stage where most of the unlabeled data can be utilized.

**Comparison with class balancing objectives.**   CPL has the effect of balancing across classes the number of unlabeled samples used to compute pseudo-labeling loss in each batch. Similar effect can be achieved by making the marginal class distribution close to a uniform distribution for each batch. We conduct such a comparative experiment by directly adding an additional objective to FixMatch: $\mathcal{L}_b = \sum_c q_c \log(q_c/\hat{p}_c)$ [22], where $\hat{p}_c$ is the mean predicted probability of class $c$ across all samples in the batch, and $q$ is a uniform distribution: $q_c = 1/C$. The error rate of adding such an objective is 7.16% on the CIFAR-10 40-label split (compared with FixMatch 7.47%±0.28 and FlexMatch 4.97%±0.06). While this approach requires instances of each class within each batch to be balanced to make sense, CPL does not have such a constraint. It is more flexible and involves less human intervention to adjust thresholds than adjusting model's predictions.

## 5   Related Work

Pseudo-Labeling [4] is a pioneer SSL method that uses hard artificial labels converted from model predictions. A confidence-based strategy was used in [6] along with pseudo labeling so that the unlabeled data are used only when the predictions are sufficiently confident. Such confidence-based thresholding also presents in recently proposed UDA [11] and FixMatch [14] with the difference being that UDA used sharpened 'soft' pseudo labels with a temperature whereas Fixmatch adopted one-hot 'hard' labels. The success of UDA and FixMatch, however, relies heavily on the usage of strong data augmentations to improve the consistency regularization. ReMixMatch [13] also leveraged such strong augmentations.

The combination of curriculum learning and semi-supervised learning is popular in recent years [37–39]. For multi-model image classification task, [37] optimized the learning process of unlabeled images by judging their reliability and discriminability. In [38], the easy image-level properties are learned first and then used to facilitate segmentation via constrained CNNs. Curriculum learning is also used to alleviate out-of-distribution problems by picking up in-distribution samples from unlabeled data according to the out-of-distribution scores [39].

Several researches have investigated on dynamic threshold in related fields such as sentiment analysis [40] and semantic segmentation [41]. In [40], the threshold was gradually reduced to make high-quality data selected into labeled data set in the early stage and large-quantity in the later stage. An extra classifier is added to automate the threshold to deal with domain inconsistency in [41]. [42] introduced curriculum learning to self-training with a steadily increasing threshold and achieved near state-of-the-art results.

## 6   Conclusion and Future Work

In this paper, we introduce Curriculum Pseudo Labeling (CPL), a curriculum learning approach of leveraging unlabeled data for SSL. CPL dramatically improves the performance and convergence speed of SSL algorithms that involve thresholds while being extremely simple and almost cost-free. FlexMatch, our improved algorithm of FixMatch, achieves state-of-the-art performance on a variety of SSL benchmarks. In future work, we would like to improve our method under the long-tail scenario where the unlabeled data belonging to each class are extremely unbalanced.

## Broader Impact

CPL fills the gap that no modern SSL algorithm considers the inherent learning difficulties of different classes during the training, and shows that by doing so, the convergence speed and final accuracy can both be improved. We hope that CPL can attract more future attention to explore the effectiveness of utilizing unlabeled data according to the model's learning status as well as the per-class learning difficulty.

## Funding Disclosure

Funding in direct support of this work: computing resource granted by Tokyo Institute of Technology and Microsoft Research Asia. This work was partially supported by Toray Science Foundation.

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
