# Appendix: FlexMatch: Boosting Semi-Supervised Learning with Curriculum Pseudo Labeling

**Bowen Zhang**[*]
Tokyo Institute of Technology
bowen.z.ab@m.titech.ac.jp

**Yidong Wang**[*]
Tokyo Institute of Technology
wang.y.ca@m.titech.ac.jp

**Wenxin Hou**
Microsoft
wenxinhou@microsoft.com

**Hao Wu**
Tokyo Institute of Technology
wu.h.aj@m.titech.ac.jp

**Jindong Wang**[†]
Microsoft Research Asia
jindwang@microsoft.com

**Manabu Okumura**[†]
Tokyo Institute of Technology
oku@pi.titech.ac.jp

**Takahiro Shinozaki**[†]
Tokyo Institute of Technology
shinot@ict.e.titech.ac.jp

## 1 Experimental Results

### 1.1 Hyperparameter setting

For reproduction, we show the detailed hyperparameter setting for each method in Table 1 and 2, for algorithm-dependent and algorithm-independent hyperparameters, respectively.

Table 1: Algorithm dependent parameters.

| Algorithm | PL (Flex-PL) | UDA (Flex-UDA) | FixMatch (FlexMatch) |
|---|---|---|---|
| Unlabeled Data to Labeled Data Ratio (CIFAR-10/100, STL-10, SVHN) | 1 | 7 | 7 |
| Unlabeled Data to Labeled Data Ratio (ImageNet) | - | - | 1 |
| Pre-defined Threshold (CIFAR-10/100, STL-10, SVHN) | 0.95 | 0.8 | 0.95 |
| Pre-defined Threshold (ImageNet) | - | - | 0.7 |
| Temperature | - | 0.5 | - |

Table 2: Algorithm independent parameters.

| Dataset | CIFAR-10 | CIFAR-100 | STL-10 | SVHN | ImageNet |
|---|---|---|---|---|---|
| Model | WRN-28-2 [1] | WRN-28-8 | WRN-37-2 [2] | WRN-28-2 | ResNet-50 [3] |
| Weight Decay | 5e-4 | 1e-3 | 5e-4 | 5e-4 | 3e-4 |
| Batch Size | 64 | | | | 128 |
| Learning Rate | 0.03 | | | | |
| SGD Momentum | 0.9 | | | | |
| EMA Momentum | 0.999 | | | | |
| Unsupervised Loss Weight | 1 | | | | |

### 1.2 Class-wise accuracy improvement.

As introduced in the paper, CPL has its ability of improving performance on those hard-to-learn classes by taking into consider the model's learning status. A detailed class-wise accuracy comparison

---

[*]Equal contribution.
[†]Corresponding author.

35th Conference on Neural Information Processing Systems (NeurIPS 2021).

is listed in Table 3, where the final accuracies of class 2, 3 and 5 with originally bad performance are improved.

Table 3: Class-wise accuracy comparison on CIFAR-10 40-label split.

| Class Number | 0 | 1 | 2 | 3 | 4 | 5 | 6 | 7 | 8 | 9 |
|---|---|---|---|---|---|---|---|---|---|---|
| FixMatch | 0.964 | 0.982 | **0.697** | 0.852 | 0.974 | 0.890 | 0.987 | 0.970 | 0.982 | 0.981 |
| FlexMatch | 0.967 | 0.980 | **0.921** | 0.866 | 0.957 | 0.883 | 0.988 | 0.975 | 0.982 | 0.968 |

## 1.3 Median error rates

We also report the median error rates of the last 20 checkpoints by allowing all methods to run the same iterations, following existing work [4]. There are 1000 iterations between every two checkpoints. The results in Table 4 show that our CPL method can dramatically improve the performance of existing SSL algorithms and the FlexMatch achieves the best accuracy. These conclusions are in consistency with the results of Table1 in the main text, showing the effectiveness of our proposed CPL algorithm.

Table 4: Median error rates of the last 20 checkpoints.

| Dataset | CIFAR-10 | | | CIFAR-100 | | | STL-10 | | | SVHN | |
|---|---|---|---|---|---|---|---|---|---|---|---|
| Label Amount | 40 | 250 | 4000 | 400 | 2500 | 10000 | 40 | 250 | 1000 | 40 | 1000 |
| PL | $77.42_{\pm1.19}$ | $48.33_{\pm2.43}$ | $15.64_{\pm0.29}$ | $90.01_{\pm0.21}$ | $58.38_{\pm0.42}$ | $37.64_{\pm0.16}$ | $\mathbf{76.44}_{\pm0.67}$ | $56.90_{\pm2.32}$ | $33.57_{\pm0.40}$ | $69.05_{\pm6.77}$ | $\mathbf{9.99}_{\pm0.35}$ |
| Flex-PL | $\mathbf{76.09}_{\pm2.25}$ | $\mathbf{47.53}_{\pm2.25}$ | $\mathbf{15.30}_{\pm0.24}$ | $\mathbf{86.60}_{\pm0.48}$ | $\mathbf{56.72}_{\pm0.54}$ | $\mathbf{36.20}_{\pm0.20}$ | $76.84_{\pm1.04}$ | $\mathbf{53.71}_{\pm2.69}$ | $\mathbf{33.19}_{\pm0.25}$ | $\mathbf{67.20}_{\pm3.99}$ | $15.10_{\pm1.33}$ |
| UDA | $10.96_{\pm3.68}$ | $5.46_{\pm0.07}$ | $4.60_{\pm0.05}$ | $\mathbf{51.97}_{\pm1.38}$ | $29.92_{\pm0.35}$ | $23.64_{\pm0.33}$ | $41.11_{\pm5.21}$ | $10.74_{\pm1.39}$ | $8.00_{\pm0.58}$ | $5.31_{\pm4.39}$ | $1.97_{\pm0.04}$ |
| Flex-UDA | $\mathbf{5.77}_{\pm0.52}$ | $5.48_{\pm0.33}$ | $\mathbf{4.52}_{\pm0.07}$ | $59.51_{\pm2.70}$ | $\mathbf{29.33}_{\pm0.23}$ | $\mathbf{23.38}_{\pm0.19}$ | $\mathbf{61.16}_{\pm4.34}$ | $\mathbf{10.88}_{\pm0.54}$ | $\mathbf{7.16}_{\pm0.20}$ | $\mathbf{6.21}_{\pm2.84}$ | $2.13_{\pm0.09}$ |
| FixMatch | $7.99_{\pm0.59}$ | $5.12_{\pm0.33}$ | $\mathbf{4.46}_{\pm0.11}$ | $48.95_{\pm1.19}$ | $29.19_{\pm0.25}$ | $23.06_{\pm0.12}$ | $44.70_{\pm6.58}$ | $12.34_{\pm2.13}$ | $7.38_{\pm0.26}$ | $\mathbf{3.92}_{\pm1.18}$ | $\mathbf{2.06}_{\pm0.01}$ |
| FlexMatch | $\mathbf{5.19}_{\pm0.05}$ | $5.33_{\pm0.12}$ | $4.47_{\pm0.09}$ | $\mathbf{45.91}_{\pm1.76}$ | $\mathbf{28.11}_{\pm0.20}$ | $\mathbf{23.04}_{\pm0.28}$ | $\mathbf{44.69}_{\pm7.49}$ | $\mathbf{9.27}_{\pm0.49}$ | $\mathbf{6.15}_{\pm0.25}$ | $20.81_{\pm5.26}$ | $12.90_{\pm2.68}$ |

## 1.4 Detailed results

To comprehensively evaluate the performance of all methods in a classification setting, we further report the precision, recall, f1 score and AUC (area under curve) results on CIFAR-10 dataset. As shown in Table 5, we see that in addition to the reduced error rates, CPL also has the best performance on precision, recall, F1 score, and AUC. These metrics, together with error rates (accuracy), shows the strong performance of our proposed method.

Table 5: Precision, recall, f1 score and AUC results on CIFAR-10.

| Label Amount | 40 labels | | | | 4000 labels | | | |
|---|---|---|---|---|---|---|---|---|
| Criteria | Precision | Recall | F1 Score | AUC | Precision | Recall | F1 score | AUC |
| PL | 0.2539 | 0.2552 | 0.2493 | 0.6542 | 0.8498 | 0.8509 | 0.8500 | 0.9833 |
| Flex-PL | **0.2865** | **0.2865** | **0.2663** | **0.6718** | **0.8544** | **0.8545** | **0.8542** | **0.9843** |
| UDA | 0.8759 | 0.8408 | 0.8086 | 0.9775 | 0.9557 | 0.9559 | 0.9557 | 0.9985 |
| Flex-UDA | **0.9482** | **0.9485** | **0.9482** | **0.9974** | **0.9576** | **0.9577** | **0.9576** | **0.9986** |
| Fixmatch | 0.9333 | 0.9290 | 0.9278 | 0.9910 | 0.9571 | 0.9571 | 0.9569 | **0.9984** |
| Flexmatch | **0.9506** | **0.9507** | **0.9506** | **0.9975** | **0.9580** | **0.9581** | **0.9580** | **0.9984** |

# 2 TorchSSL: A PyTorch-based SSL Codebase

The PyTorch [5] framework has gained increasing attention in the deep learning research community. However, the main existing SSL codebase [6] is based on TensorFlow. For the convenience and customizability, we re-implement and open source a PyTorch-based SSL toolbox, named *TorchSSL* [3] as shown in Figure 1. TorchSSL contains eight popular semi-supervised learning methods: II-Model [8], Pseudo-Labeling [9], VAT [10], Mean Teacher [11], MixMatch [12], ReMixMatch [13], UDA [14],

---

[3]Our toolbox is partially based on [7].

and FixMatch [4], along with our proposed method FlexMatch. Most of our implementation details are based on [6]. More importantly, in addition to the basic SSL methods and components, we implement several techniques to make the results stable under PyTorch framework. For instance, we add synchronized batch normalization [15] to avoid the performance degradation caused by multi-GPU training with small batch size, and a batch norm controller to prevent performance crashes for some algorithms, which is not officially supported in PyTorch.

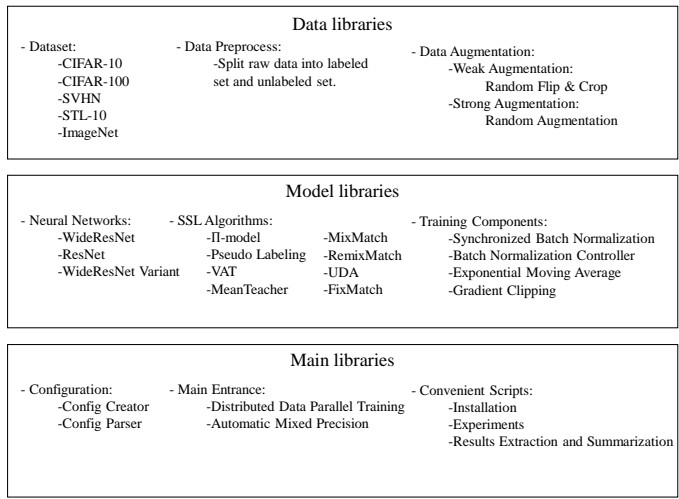

Figure 1: Components of TorchSSL.

## 2.1 BatchNorm Controller

We observed that Mean Teacher can be very unstable if we update BatchNorm for both labeled data and unlabeled data in turn. Other algorithms such as Π-Model and MixMatch also show the similar instability. Therefore, we use BatchNorm Controller to update BatchNorm only for labeled data if labeled data and unlabeled data are forwarded separately. The code of BatchNorm Controller is as follows. We record the BatchNorm statistics before the forward propagation of unlabeled data and restore them after the propagation is done.

## 2.2 Benchmark results

We comprehensively run all algorithms in our TorchSSL on four common datasets in SSL: CIFAR-10, CIFAR-100, SVHN, and STL-10, and report the best error rates in Table 6, 7, 8, and 9, respectively. These benchmark results provide a reference of using this toolbox.

Table 6: Benchmark results on CIFAR-10. The error bars are obtained from three trials.

| Algorithms | Error Rate (40 labels) | Error Rate (250 labels) | Error Rate (4000 labels) |
|---|---|---|---|
| Π-Model [8] | $74.34_{\pm1.76}$ | $46.24_{\pm1.29}$ | $13.13_{\pm0.59}$ |
| Pseudo-Labeling [9] | $74.61_{\pm0.26}$ | $46.49_{\pm2.20}$ | $15.08_{\pm0.19}$ |
| VAT [10] | $74.66_{\pm2.12}$ | $41.03_{\pm1.79}$ | $10.51_{\pm0.12}$ |
| Mean Teacher [11] | $70.09_{\pm1.60}$ | $37.46_{\pm3.30}$ | $8.10_{\pm0.21}$ |
| MixMatch [12] | $36.19_{\pm6.48}$ | $13.63_{\pm0.59}$ | $6.66_{\pm0.26}$ |
| ReMixMatch [13] | $9.88_{\pm1.03}$ | $6.30_{\pm0.05}$ | $4.84_{\pm0.01}$ |
| UDA [14] | $10.62_{\pm3.75}$ | $5.16_{\pm0.06}$ | $4.29_{\pm0.07}$ |
| FixMatch [4] | $7.47_{\pm0.28}$ | $4.86_{\pm0.05}$ | $4.21_{\pm0.08}$ |
| FlexMatch | $4.97_{\pm0.06}$ | $4.98_{\pm0.09}$ | $4.19_{\pm0.01}$ |

Table 7: Benchmark results on CIFAR-100.

| Algorithms | Error Rate (400 labels) | Error Rate (2500 labels) | Error Rate (10000 labels) |
|---|---|---|---|
| $\Pi$-Model [8] | $86.96_{\pm 0.80}$ | $58.80_{\pm 0.66}$ | $36.65_{\pm 0.00}$ |
| Pseudo-Labeling [9] | $87.45_{\pm 0.85}$ | $57.74_{\pm 0.28}$ | $36.55_{\pm 0.24}$ |
| VAT [10] | $85.20_{\pm 1.40}$ | $46.84_{\pm 0.79}$ | $32.14_{\pm 0.19}$ |
| Mean Teacher[11] | $81.11_{\pm 1.44}$ | $45.17_{\pm 1.06}$ | $31.75_{\pm 0.23}$ |
| MixMatch [12] | $67.59_{\pm 0.66}$ | $39.76_{\pm 0.48}$ | $27.78_{\pm 0.29}$ |
| ReMixMatch [13] | $42.75_{\pm 1.05}$ | $26.03_{\pm 0.35}$ | $20.02_{\pm 0.27}$ |
| UDA [14] | $46.39_{\pm 1.59}$ | $27.73_{\pm 0.21}$ | $22.49_{\pm 0.23}$ |
| FixMatch [4] | $46.42_{\pm 0.82}$ | $28.03_{\pm 0.16}$ | $22.20_{\pm 0.12}$ |
| FlexMatch | $39.94_{\pm 1.62}$ | $26.49_{\pm 0.20}$ | $21.90_{\pm 0.15}$ |

Table 8: Benchmark results on STL-10.

| Algorithms | Error Rate (40 labels) | Error Rate (250 labels) | Error Rate (1000 labels) |
|---|---|---|---|
| $\Pi$-Model [8] | $74.31_{\pm 0.85}$ | $55.13_{\pm 1.50}$ | $32.78_{\pm 0.40}$ |
| Pseudo-Labeling [9] | $74.68_{\pm 0.99}$ | $55.45_{\pm 2.43}$ | $32.64_{\pm 0.71}$ |
| VAT [10] | $74.74_{\pm 0.38}$ | $56.42_{\pm 1.97}$ | $37.95_{\pm 1.12}$ |
| Mean Teacher [11] | $71.72_{\pm 1.45}$ | $56.49_{\pm 2.75}$ | $33.90_{\pm 1.37}$ |
| MixMatch [12] | $54.93_{\pm 0.96}$ | $34.52_{\pm 0.32}$ | $21.70_{\pm 0.68}$ |
| ReMixMatch [13] | $32.12_{\pm 6.24}$ | $12.49_{\pm 1.28}$ | $6.74_{\pm 0.14}$ |
| UDA [14] | $37.42_{\pm 8.44}$ | $9.72_{\pm 1.15}$ | $6.64_{\pm 0.17}$ |
| FixMatch [4] | $35.97_{\pm 4.14}$ | $9.81_{\pm 1.04}$ | $6.25_{\pm 0.33}$ |
| FlexMatch | $29.15_{\pm 4.16}$ | $8.23_{\pm 0.39}$ | $5.77_{\pm 0.18}$ |

Table 9: Benchmark results on SVHN.

| Algorithms | Error Rate (40 labels) | Error Rate (250 labels) | Error Rate (1000 labels) |
|---|---|---|---|
| $\Pi$-Model [8] | $67.48_{\pm 0.95}$ | $13.30_{\pm 1.12}$ | $7.16_{\pm 0.11}$ |
| Pseudo-Labeling [9] | $64.61_{\pm 5.60}$ | $15.59_{\pm 0.95}$ | $9.40_{\pm 0.32}$ |
| VAT [10] | $74.75_{\pm 3.38}$ | $4.33_{\pm 0.12}$ | $4.11_{\pm 0.20}$ |
| Mean Teacher [11] | $36.09_{\pm 3.98}$ | $3.45_{\pm 0.03}$ | $3.27_{\pm 0.05}$ |
| MixMatch [12] | $30.60_{\pm 8.39}$ | $4.56_{\pm 0.32}$ | $3.69_{\pm 0.37}$ |
| ReMixMatch [13] | $24.04_{\pm 9.13}$ | $6.36_{\pm 0.22}$ | $5.16_{\pm 0.31}$ |
| UDA [14] | $5.12_{\pm 4.27}$ | $1.92_{\pm 0.05}$ | $1.89_{\pm 0.01}$ |
| FixMatch [4] | $3.81_{\pm 1.18}$ | $2.02_{\pm 0.02}$ | $1.96_{\pm 0.03}$ |
| FlexMatch | $8.19_{\pm 3.20}$ | $6.59_{\pm 2.29}$ | $6.72_{\pm 0.30}$ |