# OpenReview forum: "FlexMatch: Boosting Semi-Supervised Learning with Curriculum Pseudo Labeling"
_NeurIPS.cc/2021/Conference — NeurIPS 2021 Poster_

### Official Review · Reviewer_Lmyo · 2021-07-04

**Rating:** 7
**Confidence:** 4

**Summary:**

This work extends the popular FixMatch framework with dynamic, class-specific thresholds. Specifically, a multiplier is designed to scale the fixed threshold for each class, which is computed based on the proportion of pseudo-labeled instances with confidence values passing the threshold—the motivation is to lower the threshold for classes that have fewer pseudo-labeled instances. The effectiveness of the proposed approach, named FlexMatch, is demonstrated on CIFAR, STL-10 and ImageNet where clear improvements over FixMatch are observed.

**Limitations And Societal Impact:**

The authors have discussed the limitations in Sec.4.1. There is no obvious negative impact associated to this work.

**Main Review:**

**Strength**

- The design of dynamic thresholds is simple and seems easy-to-implement.

- The improvements brought by the dynamic thresholds are clear on CIFAR, STL-10 and ImageNet.

**Weakness**

- My biggest concern is with the novelty of this work, which seems incremental. Though the improvements look good on three datasets, the method's generality is questionable, as reflected in the poor performance on SVHN. Lines 224-230 have associated the failure on SVHN to imbalanced classes, but more concrete analysis in this regard is missing. More insights should be provided to explain and understand why the performance on SVHN is worse than FixMatch.

- What's special about TorchSSL? Does it solve any particular engineering challenges that existing code fails to cope with? This section should be moved to the appendix as it doesn't provide any new insight.

---
Post-rebuttal update:

I was initially concerned with the generality of FlexMatch due to its relatively lower performance on the simple SVHN dataset, and the paper lacking sufficient discussions in regards to how FlexMatch should be applied in practice and under which circumstances FlexMatch might perform well/poorly. These discussions are essential for practitioners since FlexMatch's underlying mechanism is hand-crafted and could possibly overfit the curated datasets. The authors' responses have addressed this concern and I suggest the authors add these discussions to the paper.

Given that the proposed approach is simple and yet effective, which is also appreciated by other reviewers, I change my rating to clear accept.

**Time Spent Reviewing:**

2.5

---

> ### Author Response · Authors · 2021-08-09
> **Response to reviewer Lmyo**
>
> Thank you for your helpful feedback. I hope these could address your concerns :)
>
> **Q1: More concrete analysis on SVHN results.**
>
> The SVHN dataset has two unique properties compared to other datasets: first, samples of each class in SVHN are imbalanced; second, the task is simple and easy (i.e., classifying digits vs. natural images in CIFAR). Due to the data imbalance issue, CPL may generate low final thresholds for the tail classes according to equation 6. Such low thresholds allow noisy pseudo-labeled samples to be trusted and learned. This is known by observing the loss descent figure where the low-threshold classes have major fluctuations. FixMatch, on the other hand, fixes its threshold at 0.95 to filter out noisy samples. Such a fixed high threshold *is not* preferable with respect to both accuracies of hard-to-learn classes and overall convergence speed as explained in our paper, but since SVHN is an easy task, the model can easily learn the task and make high-confidence predictions, it becomes less problematic to set a high-fixed threshold. The two properties together contribute to the result that FixMatch performs slightly better than FlexMatch in several cases. And we’d like to mention again that the more challenging the task is, the more significant performance improvement our method will bring, as reflected on the STL-10, CIFAR-100, and ImageNet datasets. In addition, it could be our future work to do research on imbalanced classification problems with our method.
>
> **Q2: Novelty of our method.**
>
> First of all, our targeted problem is of great importance in SSL area that the different learning status and learning difficulties of different classes *should be considered* for most SSL methods. Secondly, although our method is based on curriculum learning to flexibly adjust thresholds for different classes, which may seem quite easy, it faces several critical challenges in reality:
> - Less reliable predictions at the early stage, where the model may blindly predict most unlabeled samples into a certain class depending on the parameter initialization(i.e., more likely to have confirmation bias).
> - Threshold mapping strategy, which is non-trivial to design and will largely affect the learning process.
>
> To tackle these challenges, our proposal, FlexMatch introduces curriculum learning to dynamically adjust the learning thresholds for different classes without introducing additional computational overhead. Moreover, our method makes the following contributions:
> - Threshold warm-up, as shown in Eq. (11) to accurately estimate the learning status.
> - Non-linear mapping function, as shown in Sec. 3.3, to enable the thresholds to have a non-linear increasing curve when $\beta_t(c)$ ranges uniformly from 0 to 1.
> - Finally, our improvements v.s. naive curriculum learning (i.e., direct application of curriculum learning to select unlabeled data) are huge: e.g., on CIFAR10-4000-label split, the error rate of naive curriculum learning for data selection is 5.27 while ours is 3.95.
>
> **Q3: What's special about TorchSSL?**
>
> We use one paragraph to briefly introduce our TorchSSL in the body text because it fills the blank that currently no PyTorch-based SSL codebase is available. TorchSSL has the following characteristics.
> - There are several key engineering challenges behind. For instance, we implemented *synchronized batch norm* and *batch norm controller* to make some SSL algorithms work stably such as MeanTeacher and MixMatch, which are *not* officially supported in PyTorch.
> - In addition to these technical details, the benchmark results of TorchSSL are slightly *better* than that of the existing Tensorflow based codebase as shown in the appendix.
> - Finally, the existing codebase is not time efficient: the wall-clock time for running FixMatch is about 4 days with the Tensorflow codebase and less than 2 days with TorchSSL under the same condition and results, which means our implementation is *faster*.

---

> > ### Comment · Reviewer_Lmyo · 2021-08-11
> > **Some remaining concerns**
> >
> > Thanks for the answers. I have a few more questions.
> >
> > > Threshold warm-up, as shown in Eq. (11) to accurately estimate the learning status.
> >
> > The curriculum design is hand-crafted so its generality is a critical concern. The answer to Q1 has explained the difference in SVHN and why FlexMatch does not fare well in it. Could you expand this discussion to cover more scenarios where FlexMatch might not work well and where FlexMatch is the right place to apply.
> >
> > > There are several key engineering challenges behind. For instance, we implemented synchronized batch norm and batch norm controller to make some SSL algorithms work stably such as MeanTeacher and MixMatch, which are not officially supported in PyTorch.
> >
> > Though Sync BN is not officially supported in PyTorch. Its implementation has been open-sourced in several github repos, such as https://github.com/vacancy/Synchronized-BatchNorm-PyTorch, which are easy to find via Google.
> >
> > > Finally, the existing codebase is not time efficient.
> >
> > Could you elaborate on why specifically the existing codebase is slow and how TorchSSL solves the issue.
> >
> > So far, I still feel like TorchSSL should be briefly mentioned in the implementation part rather than being presented as a standalone section which does not provide sufficient insights.

---

> > > ### Author Response · Authors · 2021-08-11
> > > **Response to remaining concerns**
> > >
> > > Thank you again for your questions! We are glad to answer them.
> > >
> > > **Q：Generality of FlexMatch (CPL) - In what scenarios it will work.**
> > >
> > > We appreciate your comment that it is important to understand the generality of FlexMatch (CPL). Regarding this, we took deep analysis of our results (Table 1 and Figure 2) across many datasets with different situations: *task difficulty, class balance, number of labeled data, class numbers, and unseen classes*. These situations are critical in real-world applications. Though on most of the tasks of our experiments, our method can work well. We deeply analyzed these situations with the hope to understand it.
> > >
> > > Here are our conclusions:
> > >
> > > - When does FlexMatch (CPL) not work so well?
> > >
> > > When the task is **relatively simple yet unbalanced**. As shown in SVHN experiments, FlexMatch gives slightly worse results than FixMatch. But Flex-UDA still shows competitive results and Flex-Pseudo-Labeling still shows improvements, which means even in such a case where CPL is not suitable, adding it won’t do much hurt. This is because SVHN is about dight classification, which is rather simpler.
> > >
> > > - Scenarios where FlexMatch (CPL) may **bring certain/huge** improvements:
> > >     - The **more challenging** the task is; the **more classes** there are; and the **fewer labeled data** there are, the **more obvious** improvements CPL will bring. As reflected on CIFAR-10-4000-label split, the absolute improvement from FixMatch is **0.14**, and on CIFAR-10-40-label split, it is **1.79**. When it comes to CIFAR-100 (more complicated dataset with more classes compared to CIFAR-10), the improvements become **2.55** for the 10000-label split and **14.32** for the 400-label split.
> > >
> > >     - When the task is **complicated**, despite the data unbalance issue, CPL still brings major improvements. We note that although ImageNet is a rather *unbalanced* dataset (the number of images within each class ranges from 732 to 1300), applying CPL still obtains a significant performance improvement compared with existing SSL algorithms. In this situation, our motivation, i.e., different learning status of each class can become very important. All of comparison methods fail to consider learning difficulties and model’s learning status of different classes, while applying ours will boost their performance, i.e. top-1 error rate **43.08** → **35.21**; top-5 error rate **19.55** → **13.96**.
> > >
> > >     - CPL works well in more realistic situations where the unlabeled set contains **unseen** categories, such as STL-10 dataset. This is a more realistic and challenging situation, where the experimental results suggest CPL makes major improvements under such a case. E.g., for the 40-label split, the error rate is reduced from **35.42** → **10.87** by introducing CPL to the SOTA algorithm: FixMatch; and **37.31** → **12.84** by introducing to another recently proposed advanced algorithm: UDA. Such error rate reductions are impressive.
> > >
> > > | Task Difficulty (ascending) | SVHN-1k | CIFAR-10-4k | CIFAR-10-40 | CIFAR-100-10k | CIFAR-100-400 | STL-10-40 | ImageNet-1k-100k |
> > > |----------------------|---------|-------------|-------------|---------------|---------------|-----------|------------------|
> > > | Improvement over FixMatch | -0.91   | 0.14        | 1.79        | 2.55          | 14.32         | 24.55     | 7.87             |
> > > | Improvement over UDA      | 0.04    | 0.13        | 2.00        | 2.02          | 11.35         | 24.47     | -                |
> > >
> > >
> > > Finally, We'd like to mention again that apllying CPL is **cost-free** and will not only boost the accuracy but also the **convergence speed** to a large extent.
> > >
> > >
> > > **Q: Advices on TorchSSL - It’s better presented in the appendix**
> > >
> > > As we did not fully explore the Tensorflow codebase, we cannot specifically elaborate the factors contributing to its training speed. The contributor mentioned a possible reason that may lead to the slow training speed in this issue: https://github.com/google-research/fixmatch/issues/15.
> > >
> > > We agree that most of the implementations in TorchSSL are open-sourced and can be borrowed from other git repos (e.g. EMA, AMP, interleave, etc.). In this sense, TorchSSL aims to become a pytorch-friendly, easier-to-use code base by incorporating these techniques. As also mentioned in the paper, the original purpose of open-souring TorchSSL is to provide an uniform and convenient codebase for researchers working on SSL field with PyTorch. We add the tricks just to make sure the performance of other SSL methods can be correctly reproduced under PyTorch framework, as otherwise they may have very unstable training results.
> > >
> > > At this point, we would like to take your constructive advice and move this section to the appendix :)
> > >
> > > If these address your concerns, please kindly consider increasing the review score. Thank you!

---

### Official Review · Reviewer_Pkje · 2021-07-04

**Rating:** 7
**Confidence:** 4

**Summary:**

This paper introduces a curriculum learning strategy for semi-supervised learning. Their motivation is that most existing semi-supervised learning utilizes a fixed threshold to compute pseudo-labeling loss, but the fixed threshold should be adjusted according to the state of the model. Then, they propose to adjust the threshold based on the prediction of the model and design a class-specific confidence threshold. Their threshold is designed so that the class with a larger number of confident samples has a higher value of the threshold. Thus, the classes hard to learn will have a small threshold. Their way of designing the threshold is computationally efficient and will not cause much training time increase.


The experimental results show that their method is very effective in challenging situations such as when the number of labeled samples is limited or when the target dataset is challenging.

**Limitations And Societal Impact:**

They lack a section to discuss the potential social impact of their research. I do not think the research includes a particular content, which can bring negative impact, but they need to mention it somewhere in the paper.

**Main Review:**

Strength
1. Their method is simple and easy to implement. It should be important in real applications.
2. The idea behind the method is reasonable.
3. Their method is computationally efficient but provides great gains in challenging semi-supervised learning settings.
4. This paper is very easy to follow.
5. Experiments support the effectiveness of their method.

Weakness
1. The idea of introducing curriculum learning for semi-supervised learning is not very novel as done in [A].
2. Given 1, their experiments lack comparison to semi-supervised learning methods based on curriculum learning. There should be other ways of designing curriculum such as gradually increasing the confidence threshold. This paper lacks empirical and intuitive comparison to this kind of method, which makes it hard to judge the novelty of the proposed method.
3. Related to 2, the analysis of results is not enough to fully understand the method. From the intuition of the method, the expected results can be that this method improves accuracy on hard-to-classify classes. Did you observe the results?
4. The proposed threshold design will balance the number of unlabeled samples used to compute pseudo-labeling loss per class. If we have the objective to balance it such as entropy of marginal class distribution for unlabeled samples, do we have the same effect as the proposed method?

In summary, this paper provides a simple and well-motivated technique to improve semi-supervised learning based on pseudo-labeling. However, it lacks comparison to existing methods with curriculum learning. If they can provide simple baselines with some curriculum learning, the readers would understand the advantage of the proposed method better.


[A] Curriculum Labeling: Revisiting Pseudo-Labeling for Semi-Supervised Learning, AAAI 2021


---post rebuttal update---
The rebuttal addressed my concerns. Despite its simplicity, the proposed method is able to boost the performance of semi-supervised learning even for hard classes. I would like to raise my rating to 7.


**Time Spent Reviewing:**

2

---

> ### Author Response · Authors · 2021-08-09
> **Response to reviewer Pkje**
>
> Thank you for your helpful comment. I hope these could address your concerns :)
>
> **Q1: Comparison with [A].**
>
> The only similarity between our paper and this one is the **title**. However, our key technology is **significantly different** from theirs:
> - First, they use curriculum learning for **unlabeled data selection**, i.e., use the model trained on the labeled dataset to manually label unlabeled data, and then *add the pseudo-labeled data to enlarge the labeled dataset*. In contrast, we use curriculum learning to render the pseudo labels *to different classes* and *at different time steps*, which is determined by dynamic thresholds.
> - Second, the ways of using curriculum learning are also different. They **manually** pre-define the curriculum pace by using the percentile increasing from 0% to 100% with 20% increments each time. In contrast, we **automate** the thresholds in each iteration and for each class according to the model’s learning status, which means the thresholds can *increase, decrease or stay unchanged*. Therefore, our proposed CPL is more flexible and does not involve man-made predefinitions (i.e. 20% increments).
> - Third, we do **not** do select-add-and-train like they did since it would introduce extra computations (e.g. extra forward propagations when using the trained model to select), our CPL is cost-free but effective.
> - Finally, our proposed method has much **stronger** results compared to theirs and can be easily adapted to existing SSL algorithms. Specifically, their proposed methods have **worse** results than FixMatch using the same model settings (i.e. WRN-28, ResNet50), whereas ours can **substantially outperform** the current SOTA algorithm: FixMatch. For instance, on CIFAR10-4000-label split, their error rate is **5.27** while ours is **3.95**; with **500 labels**, their error rate deteriorates to about **10.50** whereas ours with **250 labels** is **4.80** and with **only 40 labels** is **4.99**.
>
> We will add a brief summary of this comparison to the related work in the next version of the paper.
>
> **Q2: Does the proposed method improve accuracy on hard-to-classify classes?**
>
> Yes, our FlexMatch method can improve the accuracy of hard-to-classify classes. As shown in Figure 3.c and Figure 3.d, the introduction of CPL raises the accuracy of hard-to-classify classes to a large extent during the first 200k iterations. And here is the final **class-wise accuracy on CIFAR-10-40-label split**:
>
> | Class No. |     0 |     1 |         2 |         3 |     4 |         5 |     6 |     7 |     8 |     9 |
> |-----------|------:|------:|----------:|----------:|------:|----------:|------:|------:|------:|------:|
> | FixMatch  | 0.961 | 0.975 | **0.785** | **0.851** | 0.972 | **0.874** | 0.985 | 0.976 | 0.978 | 0.975 |
> | FlexMatch | 0.969 | 0.984 | **0.916** | **0.863** | 0.966 | **0.883** | 0.984 | 0.976 | 0.981 | 0.973 |
>
> The accuracy of the No.2 class with the worst result in FixMatch is improved from 78.5% to 91.6% after adapting CPL.
>
> **Q3: Comparing with balanced marginal distribution objective.**
>
> We added an experiment of this on CIFAR-10-40-split, the result of adding such an objective to FixMatch is **92.84%** (i.e. 7.16 error rate), which is very close but slightly worse than the original FixMatch, hence much worse than our proposed method (4.99 error rate). Although forcing the marginal distribution to be balanced can alleviate that in the early stage most unlabeled samples tend to be classified into the same class, it introduces a man-made presupposition that **within each mini-batch** the marginal distributions of each class are also balanced, which may not be true due to the random sampling.
> On the other hand, our proposed method *does not* make such a constraint. The thresholds as well as the number of samples belonging to each class within each mini-batch are optimized according to the current learning status, and will be adjusted along with the whole training process. And in our method, the problem of blindly predicting into the same class at the early stage is alleviated by the threshold warm-up as mentioned in Section 3.2.

---

### Official Review · Reviewer_Qc4v · 2021-07-17

**Rating:** 3
**Confidence:** 4

**Summary:**

The paper proposes to incorporate ideas from curriculum learning into semi-supervised learning. Semi-supervised learning consists on training models with only a subset of labeled samples, and a typically larger set of unlabeled samples. The paper proposes to improve on ideas from pseudo-labeling whereas fixed thresholds are defined to decide the pseudo-labels for unlabeled samples as the training process goes. In this approach, the idea is to modulate the thresholds used during the training process as a function of time (iterations) and the individual classes being classified. The motivation is that some classes might be harder to learn. In order to deal with initial conditions, the paper proposes to use a warming up strategy whereby an additional class (undecided) is considered for samples that have not reached a confidence to be assigned to any class. The proposed method in combination with other methods based on consistency regularization such as FixMatch lead to improved accuracies.

**Ethical Concerns:**

Not particularly.

**Limitations And Societal Impact:**

I don't believe I saw much in the way of limitations or any discussion about this part.

**Main Review:**

Strengths:
+ Curriculum learning is well motivated in terms of the presented task and seems a good strategy to pace the thresholds used in semi-supervised learning.
+ TorchSSL is proposed as a library re-implementing prior algorithms in a unified codebase. This could potentially provide a resource for further development in this area.
+ The proposed method is general enough that it can be combined with others and in experimental results higher accuracies are reported after applying the proposed method.

Weaknesses:
+ There has been already a prior proposal to use Curriculum-based pacing for choosing the thresholds for Semi-supervised learning for pseudo-labeling: Curriculum Labeling: Revisiting Pseudo-Labeling for Semi-Supervised Learning. AAAI 2021. February 2021. The idea of changing thresholds for deciding on the unlabeled samples hence has already explored before. Without direct comparisons or discussions it diminishes the contributions of the present work. However I do acknowledge some differences: the current work seems to consider per-category adaptation of the pacing on the curriculum, whereas this other work considers a single threshold. However whether this difference is important as claimed in the present work, it seems that it would be something that needs validation.
+ There were some technical details that could only be resolved after reading the supplementary material, something important such as what is or are the neural network architectures that are being trained for obtaining the results on Table 1.
+ More importantly there are some inconsistencies reported on the number for Imagenet-1k where FixMatch is reported as having a top-5 error of 19.55% but the prior work both the original FixMatch paper, and the above referenced AAAI 2021 paper report 10.87% for FixMatch. The conditions seem the same 10% of data annotated (= 100k samples annotated). On the surface the presented method would be impressive as the reported FixMatch error is reported as 19.55% and it goes down to 13.96% -- however only if we accept a version of FixMatch that starts at a much worse level of performance this might be true. Section 4.2 is really light on the details for this part and from the Supplementary material it doesn't seem like the settings were fundamentally different than in these two prior works.
+ The authors cite [27] but don't seem to follow some of the recommendations in that work regarding the brittleness of many methods under unlabeled samples that include categories that are not part of the target set. Other recent works have tested also under this scenario but I don't seem this type of experiment in the current work.

**Time Spent Reviewing:**

4

---

> ### Author Response · Authors · 2021-08-09
> **Response to reviewer Qc4v**
>
> Thank you for the constructive feedback. I hope these could address your concerns :)
>
> **Q1：Comparison with AAAI2021 paper: Curriculum Labeling: Revisiting Pseudo-Labeling for Semi-Supervised Learning.**
>
> The only similarity between our paper and this one is the **title**. However, our key technology is **significantly different** from theirs:
> - First, they use curriculum learning for **unlabeled data selection**, i.e., use the model trained on the labeled dataset to manually label unlabeled data, and then *add the pseudo-labeled data to enlarge the labeled dataset*. In contrast, we use curriculum learning to render the pseudo labels *to different classes* and *at different time steps*, which is determined by dynamic thresholds.
> - Second, the ways of using curriculum learning are also different. They **manually** pre-define the curriculum pace by using the percentile increasing from 0% to 100% with 20% increments each time. In contrast, we **automate** the thresholds in each iteration and for each class according to the model’s learning status, which means the thresholds can *increase, decrease or stay unchanged*. Therefore, our proposed CPL is more flexible and does not involve man-made predefinitions (i.e. 20% increments).
> - Third, we do **not** do select-add-and-train like they did since it would introduce extra computations (e.g. extra forward propagations when using the trained model to select), our CPL is cost-free but effective.
> - Finally, our proposed method has much **stronger** results compared to theirs and can be easily adapted to existing SSL algorithms. Specifically, their proposed methods have **worse** results than FixMatch using the same model settings (i.e. WRN-28, ResNet50), whereas ours can **substantially outperform** the current SOTA algorithm: FixMatch. For instance, on CIFAR10-4000-label split, their error rate is **5.27** while ours is **3.95**; with **500 labels**, their error rate deteriorates to about **10.50** whereas ours with **250 labels** is **4.80** and with **only 40 labels** is **4.99**.
>
> We will add a brief summary of this comparison to the related work in the next version of the paper.
>
> **Q2: Network architectures are put to the appendix instead of the main body.**
>
> We use ResNet50 for ImageNet and WideResNet for other datasets following the existing works for a fair study. We will move the description of network architectures from the appendix to the main body in the next version of the paper.
>
> **Q3：ImageNet result inconsistency with two other works.**
>
> The inconsistency of the ImageNet result reported in this work and two other works is because of the different experiment settings, in particular, **labeled data amount, learning rate, and batch size**. In the AAAI2021 and the FixMatch paper, they use **10%** labeled data whereas we use **100k**. Due to the fact that ImageNet contains more than 1M images, e.g. ImageNet2012 has over 1.28M thus 10% being **128k** labels, our labeled data amount is **smaller** than theirs. Besides, they both use a learning rate of **0.1**, whereas we use the learning rate of **0.03** to make it consistent with the experiments on other datasets. While the learning rate being smaller, the number of training iterations remains the same, which also causes our reported results lower than theirs. As for the batch size, FixMatch uses **1024** and we use **32** due to the computational resource limitation. All these factors together contribute to the inconsistency of the results. These parameter settings are introduced in Table 4 in our appendix, as well as in their papers. However, despite the hyperparameter differences, **the comparison between FixMatch and FlexMatch within our codebase under the same condition is still fair**, and the performance improvement is noticeable. We will add more ImageNet results in the next version of our paper to address further concerns.
>
> **Q4：Evaluation under the case where unlabeled samples include categories that are not presented in the target set.**
>
> We did evaluate our method and other baselines under the scenario where unlabeled samples include unseen categories, i.e., on the STL-10 dataset. As written in the official introduction of this dataset, *‘These examples (unlabeled samples) are extracted from a similar but broader distribution of images. For instance, it contains other types of animals (bears, rabbits, etc.) and vehicles (trains, buses, etc.) in addition to the ones in the labeled set.’* This is just the case where unlabeled samples include unseen classes as reviewer posted. Therefore, our experiments on the STL-10 dataset are under this case, where our FlexMatch **outperforms** other baselines, as indicated in L215-L223 in the main paper.

---

> > ### Author Response · Authors · 2021-08-14
> > **we hope our response can resolve your concerns**
> >
> > If you have any further concerns, feel free to ask at any time, we are very glad to answer.
> > And if we addressed your concerns, please kindly consider increasing the review score. Thank you very much.

---

> > > ### Comment · Reviewer_Qc4v · 2021-08-14
> > > **Issues with experimental settings**
> > >
> > > Concerns regarding comparison to prior work have been addressed to some extent except on the claims about performance which is related to my second concern: The inconsistency of Imagenet results with respect to what is reported in prior works. It just doesn't seem justified after the explanation provided -- there's no justification for using different settings -- e.g. 100k vs 128k amount of labels or learning rate of 0.1 vs 0.03. The fact that a proposed method does better under a specific set of hyperparameters is not enough to conclude relative advantages of different methods. e.g. If Method A works best under learning rates of 0.03 and Method B works best with learning rates of 0.1, then the methods should be trained each under the hyperparameter settings that are optimal for each method. This is fair. The performance gap is large and is unclear where this new proposed method stands if the experiments were conducted in the way is suggested here.

---

> > > > ### Author Response · Authors · 2021-08-14
> > > > **Response to the additional issue**
> > > >
> > > > We are very confused that you lowered the score from 4 to 3 while admitting that we *did* address your 3/4 concerns, i.e., novelty, network architecture, and test dataset.
> > > >
> > > > As for your remaining concern about ImageNet experiments:
> > > >
> > > > - First of all, ImageNet is *only* one of our dataset (others includiing CIFAR-10/100, STL-10, SVHN). And the results in Table 1 have already shown our effectiveness, as you and other reviewers acknowledged. For instance, Mixmatch and Remixmatch even did not run on ImageNet.
> > > >
> > > > - Secondly, *please do not misunderstand*: we **did not** choose this experiment setting because it was advantageous for us.
> > > >
> > > >     - In fact, the setting for ImageNet in FixMatch paper used several special settings that may bring further advantages: they used a different initial learning rate with a different schedule, a different unlabeled loss weight, confidence threshold, labeled-to-unlabeled data ratio as well as a very *large* batch size. We did not make these special hyper-parameter tweaks but simply used the same hyper-parameters as for the other datasets which follow exactly the FixMatch baseline. Plus, Fixmatch cannot be easily reproduced with normal computing resources since it used a TPU device with 32 cores.
> > > >
> > > >     - So, *it is not the case* that we deliberately adjust the parameter settings so that our proposed method can outperform existing methods. In fact, even with the same hyper-parameters reported in the FixMatch paper, our method still shows obvious superiority: We added such an experiment following their hyper-parameter settings with the only difference being that their batch size is 1024 and ours is 32 -- **due to the computational resource limitation, we do not have the ability to reproduce an experiment of a batch size of 1024 as Google did**. So far, after 100 out of the total 3000 training epochs, the top-1 accuracy for FixMatch vs our FlexMatch is **25.06 vs 35.06**, and **48.11 vs 59.78** for the top-5. It suggests the improvement might be even larger under such an experiment setting. The final result will be added to the paper once it’s obtained.
> > > >
> > > > - Finally, **we emphasize again that the hyper-parameter difference is not meant to bias towards any method.** We are very confident that our proposed method can show significant improvement against the FixMatch baseline on the ImageNet dataset, as it's true on other challenging benchmark datasets (CIFAR-100 and STL-10), *under all circumstances as long as the experimental conditions being the same* (which is true in our reported comparison). And our convergence is better.
> > > >
> > > > This table shows the top-5 accuracy comparison on ImageNet under different experimental conditions, the latter experiment is currently ongoing.
> > > >
> > > > | Experiment Settings | Use Ours                    | Follow FixMatch             |
> > > > |---------------------|-----------------------------|-----------------------------|
> > > > | FixMatch            | 80.45 (at 2^20 iterations)     | 48.11 (at 100/3000 epoch)     |
> > > > | FlexMatch           | **86.04** (at 2^20 iterations) | **59.78** (at 100/3000 epoch) |
> > > >
> > > >
> > > > Therefore, with all due respect, we cannot accept a score of 3 as it is neither reasonable nor fair-minded. We sincerely hope you could reconsider. Thank you!

---

> > > > > ### Author Response · Authors · 2021-08-17
> > > > > **We hope the response solves your additional concerns**
> > > > >
> > > > > We made a response showing that the experimental settings are not biased towards any method, and our proposed method can also surpass the baseline system to a large extent under their hyper-parameter settings. If these resolve your additional concerns on the ImageNet experiment, please kindly consider increasing the rating. Thank you.

---

### Official Review · Reviewer_w5Zg · 2021-07-17

**Rating:** 7
**Confidence:** 4

**Summary:**

The authors proposed a simple and straightforward method to address an obvious but critical and long-lasting issue in semi-supervised learning. They showed a number of analyses and demonstrated superior performance over state-of-the-art methods (mostly) consistently on five datasets. The proposed method is simple and easy to use without extra computation needed. Most importantly, I do see the possibilities of adapting this method widely across semi-supervised learning or self-training methods that use pseudo-labeling. I would recommend acceptance at this time.


**Ethical Concerns:**

As with any other papers for semi-supervised learning, we might have a risk of introducing biases, because we are leveraging a smaller number of samples that are labeled. These samples could already be biased. However, all recent semi-supervised learning methods are striving to address this issue.

**Limitations And Societal Impact:**

The authors did analyze the failed cases and identify the potential limitation of the proposed method (L224 - L230).

**Main Review:**


### Strengths: Describe the strengths of the work. Typical criteria include: soundness of the claims (theoretical grounding, empirical evaluation), significance and novelty of the contribution, and relevance to the NeurIPS community.

- Paper is well-written through entire paper and super easy to follow
- Proposed method is extremely simple without adding extra computation
- Plan to public release TorchSSL, a PyTorch library for semi-supervised learning
- Providing two versions of evaluation metrics for fair comparisons with the existing method and argue that choosing the best error rates might be more suitable for future research because the convergence speed is also important and won't be able to be reflected in the median error rate of the last 20 checkpoints.
- thorough experiments with 5 datasets and analysis on convergence speed
- analyze the failed cases and identify the potential limitation of the proposed method (L224 - L230)

### Weaknesses: Explain the limitations of this work along the same axes as above.
- Interestingly, I don't think I can find any obvious weaknesses. This is a well-written paper.

### Additional feedback, comments, suggestions for improvement and questions for the authors:
- L202: "When the amount" --> "when the amount"

**Time Spent Reviewing:**

4

---

> ### Author Response · Authors · 2021-08-09
> **Response to reviewer w5Zg**
>
> Thank you for your careful and positive feedback, and even kindly pointing out the typo error! We will fix it :)

---

> ### Public Comment · ~Yuxin_Guo2 · 2023-08-09
> **1**
>
> 2

---

### Decision · Program_Chairs · 2021-09-27

**Decision:**

Accept (Poster)

**Comment:**

This paper proposes Curriculum Pseudo Labeling (CPL), an extension to semi-supervised learning algorithms that uses an estimate of how well the model has learned each class to set per-class thresholds for promoting unlabeled data to pseudolabeled data. This conceptually simple idea has a significant impact in practice. It is also simple to implement and does not significantly add to the computational cost of the base SSL methods. (It actually can speed up convergence significantly.) Further, the combination of CPL and FixMatch (called FlexMatch) sets new state-of-the-art scores on several SSL benchmarks. Finally, the paper includes the release of Torch SSL, a codebase for the study of SSL.

The reviewers generally liked that the method was conceptually simple, effective, and easy to combine with many existing methods. Three of the four reviewers clearly favored acceptance, for the above strengths. Some of the concerns they had initially were addressed during the discussion period, and the authors are encouraged to include these clarifications in the final version of the paper. One reviewer had serious concerns about the experimental setup. The results reported in the paper for baselines differ from some previously reported in the literature. The authors attribute this to both a difference in the number of labeled examples (which is fair to vary in SSL experiments) and differences in hyperparameters. The authors are encouraged to both clarify whether the hyperparameters were tuned to be optimal for each baseline and also to add comparisons on the same number of labeled examples as previously used in the literature (perhaps in an appendix) for a more detailed comparison.